# Neuropsychological Stimulation Program for Children from Low Socioeconomic Backgrounds: Study Protocol for a Randomized Controlled Trial

**DOI:** 10.3390/healthcare12050596

**Published:** 2024-03-06

**Authors:** Pablo Rodríguez-Prieto, Ian Craig Simpson, Diego Gomez-Baya, Claudia García de la Cadena, Desirée Ruiz-Aranda, Joaquín A. Ibáñez-Alfonso

**Affiliations:** 1Human Neuroscience Lab, Department of Psychology, Universidad Loyola Andalucía, Avda. de las Universidades s/n, 41704 Dos Hermanas, Spain; prprieto@uloyola.es (P.R.-P.); daruiz@uloyola.es (D.R.-A.); 2ETEA Foundation, Development Institute of Universidad Loyola Andalucía, Calle Escritor Castilla Aguayo n°4, 14004 Córdoba, Spain; 3Department of Experimental Psychology, Campus Universitario de Cartuja, University of Granada, 18071 Granada, Spain; isimpson@ugr.es; 4Department of Social, Developmental and Educational Psychology, Campus de El Carmen, University of Huelva, 21071 Huelva, Spain; diego.gomez@dpee.uhu.es; 5Department of Neuropsychology, Universidad del Valle de Guatemala, Calle 11, Zona 15, Post n° 82, Guatemala City 01901, Guatemala; claudigd@uvg.edu.gt

**Keywords:** neuropsychology, childhood development, vulnerability, cognitive stimulation, emotional, randomized controlled trials

## Abstract

Background: Guatemala remains one of the poorest countries in Central America and suffers from high rates of social inequality and violence. In addition to the negative impact that two years without attending school has had on Guatemalan children due to the consequences of the COVID-19 pandemic, this unfavourable socioeconomic context poses a risk to children’s emotional and cognitive development. This work presents a protocol for implementing a cognitive and emotional stimulation program aimed at increasing the academic performance of these children and consequently improving their quality of life. Methods: The protocol proposes the implementation of a randomized controlled trial to assess the efficacy of a 24-session-long stimulation program. It targets the cognitive functions of attention, language, executive functions, and social cognition, using the digital neurorehabilitation platform NeuronUP. The participants (*n* = 480) will be randomly assigned to an Experimental or Control group. Pre- and post-intervention assessments will be carried out, together with a follow-up in the next academic year, in which both groups will change roles. Results will be compared for the first and second years, looking for differences in academic and cognitive performance between groups. Discussion: Mid- and long-term outcomes are still unknown, but effective interventions based on this protocol are expected to facilitate the following benefits for participants: (1) improved cognitive and emotional development; (2) improved academic performance; (3) improved well-being. We expect to create a validated neuropsychological stimulation program that could be applied in similar socioeconomically disadvantaged contexts around the world to help these children improve their life chances.

## 1. Introduction

The pandemic (COVID-19) represented an extreme threat to the advancement of education in Guatemala because of the significant impact it has had at various levels. At a local level, educational centres were closed for almost two years, and more broadly, there has been an economic recession at the national and worldwide levels. The closures within the school system caused a stagnation of learning, along with an increase in school dropouts and a reduction in school attachment [1]. The country’s economic crisis impacted the most vulnerable individuals, directly affecting their quality of life and increasing the inequalities that already exist within the country [2]. Some of the impacts on students and their families, who have been in a forced distance-learning model for the past two years, include worsening nutritional situations (due to the students not being fed at schools), deteriorating mental health (particularly increased anxiety, stress, and burnout symptoms [3,4,5,6]), and a worsening vulnerablity [1]. It is important to mention that child labour, abuse, and unwanted pregnancies are problems that also manifest when schooling becomes home-based [7,8]. Consequently, it is necessary to implement concrete strategies to counteract or lessen the impact of this situation as much as possible, and this is the motivation for developing the following intervention protocol.

Guatemala remains one of the poorest countries in the American continent with high rates of social inequality and violence, in which structural problems such as racial discrimination, gender inequality, extreme poverty and exclusion, an unstable political situation, and an aberrant lack of access to justice constitute obstacles to the full enjoyment of human rights [2]. Additionally, racism and discrimination against indigenous peoples are deeply rooted. The indigenous population faces more unfavourable conditions than the non-indigenous population in terms of access to employment, health, education, as well as average income and rates of extreme poverty [2]. Guatemala is a diverse, multilingual, and multicultural country with 22 language communities. According to data from the last Census [9], the estimated Guatemalan population in 2020 was 16,858,333 inhabitants. The country’s gross domestic product per capita is around 39,043 quetzals, which would correspond to approximately 5 US dollars. It is the country in Latin America with the most chronic child malnutrition and the fourth worldwide, according to data from the World Bank [2]. Of relevance, in terms of the distribution of the country’s inhabitants, half of the population resides in rural areas, where three out of four inhabitants live in poverty [2]. The schools where the intervention will be carried out are in so-called high-risk areas located within the department of Guatemala. These are areas where extreme violence, assaults, murders, and extortion are situations that happen daily, and the risk of dropping out of school is very high due to reasons including lack of money, child labour, or student demotivation [10]. 

Living conditions during childhood and adolescence affect the cognitive and emotional development of minors, resulting in important long-term effects in their adult life [11,12,13,14]. Existing literature on growing up in disadvantaged environments reports that language development, attention, and executive functions can all be affected. Additionally, experiencing these types of adverse contexts can also have a considerable negative impact on children’s social skills and emotional regulation [15,16,17,18]. The development of this study protocol arises from the need to respond to some of the problems discovered after the conclusion of a previous project [19]. Its goal was to improve existing neuropsychological assessment task scores for children and adolescents in Guatemala and study how different factors within their environment could affect their cognitive development. The results obtained in this previous study revealed that Guatemalan children and adolescents participating in the project needed effective support on multiple levels due to their situations of vulnerability. In this study, vulnerability was defined as belonging to an especially disadvantaged socioeconomic context, both in terms of poverty and direct exposure to extreme violence; this was the definition used in the previous study [20]. This showed that the hypotheses regarding the negative impact of growing up in disadvantaged socioeconomic environments (e.g., low educational level of caregivers, little intellectual stimulation in their environment, food insecurity) on the cognitive performance of children and adolescents were confirmed in this population [20]. Additionally, it was possible to detect a series of mental health difficulties in some of the participants that could not be left unattended. This included high levels of anxiety and depression, especially among children from urban areas with high rates of violence [21]. In summary, evidence indicates that living conditions during childhood and adolescence affect the cognitive and emotional development of children, having significant effects on their academic performance, which in turn tends to condition their long-term adult life [12,13,14]. Likewise, the stress associated with poverty causes a serious impact on the emotional and social development of these children, highlighting the urgent need to work in these areas [22,23]. 

Neuropsychological stimulation programs focused on both cognitive and emotional functions have a long tradition in the clinical field, being the treatment of first choice in various neurodevelopmental disorders [24]. These programs are designed to train specific cognitive skills (such as attention, memory, language, etc.), with a wide range of activities, games, and other media, so patients can improve cognitive skills that are not fully acquired/developed (due to neurodevelopmental disorders) or have declined (due to acquired brain damage or degenerative illnesses). In addition, research exists demonstrating the efficacy of neuropsychological interventions focused on specific cognitive domains in non-clinical child populations belonging to socioeconomically vulnerable contexts [25,26,27]. The results suggest that protocols employing different agents and materials in cognitive intervention, with early intervention and a large number of sessions, are the most effective. It has also been shown that children living in socioeconomically vulnerable contexts can also benefit from learning emotional regulation tools to constructively face the various risk situations they are frequently exposed to in their daily lives [28]. Nevertheless, we noted that scientific evidence on the efficacy of neuropsychological stimulation programs for children at risk of social exclusion, and their positive effects on cognitive and emotional skills, as well as their expected impact on academic performance and quality of life, is still somewhat limited [27,29,30].

The previous studies and factors mentioned above have been strong motivation for the development of the following protocol for the neuropsychological stimulation program. The implementation of a neuropsychological intervention program that stimulates both cognitive development (especially those mental functions in which a greater negative impact of the socioeconomic context was detected), as well as emotional skills that reinforce the socio-emotional development of these children (an important protective factor against mood swings and other mental health disorders to which they are frequently exposed) would be greatly beneficial for children living in socioeconomically vulnerable contexts [31,32].

## 2. Materials and Methods

### 2.1. Participants and Study Setting

#### 2.1.1. Participants’ Characteristics

Students from the 5th grade of primary school, from educational centres located in particularly vulnerable areas from Guatemala City’s suburbs, will be invited to participate in the intervention program. The reason for choosing this grade is that the approximate age of the children in this school stage are at the minimum that ensures they have acquired certain skills necessary for adequate participation in the intervention program (e.g., sufficient ability to sustain their attention, reading comprehension, etc.), and therefore, they may benefit more from the proposed intervention. It has been found that this is the optimal educational level for stimulation programs to succeed [33,34]. Most children will be between 10 and 12 years old. However, given the challenging social background and course disruption caused by the pandemic, it is likely that there will be children 1–2 years older in this grade. The following additional inclusion criteria will be considered:-Being a proficient speaker of the Spanish language.-Having acquired sufficient reading skills to understand the activities included in the intervention (percentile score > 5 in the Inter-American Reading Series L-3-DEs).-Having a sufficient level of non-verbal intelligence to understand and interact with the activities included in the intervention (percentile score > 5 in the Non-Verbal Intelligence Test TONI-2).-Not having a diagnosis of serious sensory or motor difficulties, or having insufficiently corrected ones, which would prevent adequate performance in intervention activities.

#### 2.1.2. Recruitment Strategy

Fe y Alegría is an international Jesuit NGO dedicated to creating schools and offering education in socioeconomically disadvantaged areas throughout many Spanish-speaking countries. In Guatemala, they currently have a country-wide network of educational centres, consisting of 46 schools in total. Five centres located in the urban area have been chosen as a sample to represent the total population of their students due to the large logistical challenge of accessing more remote schools and because they have sufficient technical infrastructure (such as adequate Wi-Fi network stability) to carry out the project. Three of the centres are in the suburbs of the Department of Guatemala, and two are in the Municipality of Villa Nueva. All of them are in colonias (neighbourhoods) characterized by low socioeconomic level, aggravated by very high exposure to violence; assaults, deaths, and extortions are daily situations in these communities, which makes them especially vulnerable areas. Between these five centres, there are approximately 1400 5th grade children enrolled. Every child have to provide oral consent to participate in this study, and their legal guardians must sign an informed consent document allowing the use of collected data for research purposes, after being extensively briefed about the goals, methods, and potential benefits of participation (this document was provided by the research team and can be seen in the Appendix A). 

#### 2.1.3. Sample Size

To ensure that the size of the improvements realized by the intervention are of practical significance, we propose a minimum effect size of Cohen’s *d* = 0.3, which is a small-to-medium effect. A power calculation using this effect size and the proposed analysis plan (see below) was carried out using the program G*Power [35] and indicated that the minimum sample size we require for the study is n = 352. However, due to the difficult living conditions in the proposed recruitment areas (gang violence, long distances from residences to school centres, etc.), we expect a high dropout rate. Therefore, we propose to recruit a total of 480 children as this would allow for slightly more than 25% of the sample to stop participation in the study yet still allow us to maintain sufficient statistical power to detect the target effect size. Furthermore, the figure of 480 represents approximately one-third of 5th grade students attending Fe y Alegría Guatemala across their 46 centres. Thus, we would only need consent from one in every three children to achieve the necessary sample size. This sample size could be reduced if research teams were prepared to look for larger effect sizes. Furthermore, once this protocol is implemented, research groups should consider creating a contingency plan based on the specific conditions they encounter to address potential difficulties in recruitment.

### 2.2. Objectives and Hypothesis

The main objective of this study protocol is to improve the cognitive performance and emotional competence of students at risk of social exclusion in vulnerable areas of Guatemala (due to their high exposure to violence and low socioeconomic level) and consequently improve their academic performance, psychological adjustment, and quality of life. The specific objective is the development of a comprehensive neuropsychological stimulation program in which the following cognitive and emotional functions are targeted: attention, language, executive functions, social cognition, and emotional competence. This comprehensive intervention responds to the needs identified in our previous studies, and we hypothesize that strengthening cognitive abilities and emotional competence will improve overall health and well-being, as previous studies have shown [25,26,27].

### 2.3. Instruments

#### 2.3.1. Comprehensive Assessment Battery

To obtain the baseline data of the participants and evaluate their subsequent cognitive and emotional performance after the intervention, we will use a battery of neuropsychological tests to assess the cognitive domains of language [36,37,38]: attention [39], executive functions [40,41,42,43,44], and social cognition [45]. We will also assess their mental health [46,47], emotional management [48], and quality of life [49]. Lastly, we will take measurements of their family’s socioeconomic level, clinical history, food consumption [50], exposure to violence [51], and academic performance (for detailed explanations of the assessment instruments, see Appendix A). In total, there will be four assessment sessions, each lasting between 45 and 60 min. The session plan has been designed in a modular way that allows for flexible administration in order to accommodate participants who cannot attend the educational centre on a daily basis. Each module can be assessed in any order, and they can be applied one after the other within the same session (providing breaks in between), although assessing each module on different days would be ideal. Given the large number of tests that will be required for a comprehensive assessment, it will become difficult to compare results across tests. For this reason, converting results to percentiles or z-scores will need to be performed in order to make meaningful comparisons between instruments.

#### 2.3.2. Cognitive and Emotional Intervention Program

The intervention program will be carried out using tablets with access to the digital neurorehabilitation platform NeuronUP [52], a tool for neuropsychological rehabilitation, stimulation, and session management tool [53,54,55]. Computerized neuropsychological interventions are an alternative to the classic pencil and paper tasks that offer patients a novel and entertaining way to train their cognitive skills. These interventions have been internationally validated [56,57,58,59,60,61]. The sessions themselves will be divided into 30 min of stimulation time and 15 min for distributing the tablets at the beginning, logging in to the students’ accounts, and collecting them at the end. Since the stimulation program will be carried out through the use of this app, there are no associated health risks for the participants; it is a non-invasive method of cognitive stimulation prepared to promote participant engagement and reduce frustration by presenting exercises as mini-games. The sessions will be programmed by specialists in neuropsychology from the research team using the NeuronUP2GO functionality, an online application that allows for managing the stimulation sessions of users located anywhere around the world. Remote monitoring provides continuous control, thanks to the fact that all activities and results are stored on the platform itself. In addition, using NeuronUP2GO, professionals can adapt sessions based on the progress of each user, thus achieving a controlled and personalized intervention. The stimulation program will aim to train the participants in activities targeting attention, language, executive functions, and social cognition in a cyclical manner, dedicating each day to a specific domain. Activities will be different in every session to avoid boredom. However, they will be training the same cognitive domains, so participants are able to work on improving them at a similar pace. The program will also have three levels of difficulty (easy, medium, and hard) that will increase in difficult every two cycles, so the students have time to adapt to their current difficulty level without being rushed. There will be two activities for each cognitive skill at every difficulty level (see Figure 1). 

This also makes it possible to apply the program in a flexible way to entire class groups in which there are usually small differences in age as well as different rates of development and special needs. Additionally, it allows children who take longer to adapt the time they need to develop the intended capabilities gradually, while providing every child the same level of stimulation, so the whole class can keep steady progress. The administration of tablets to participants in each session will be carried out by personnel, duly trained and appointed by the management team of the educational centres (normally academic tutors or those responsible for the computer area), with periodic supervision by neuropsychology and educational psychology specialists from the project in case any problems or questions arise. The first session will include an initial presentation in which professionals instruct participants on how to operate the tablets and the app, making it slightly longer than other sessions. A pilot trial has already been carried out with Guatemalan children from these schools to adapt the activities culturally and linguistically. A team of Guatemalan and Spanish neuropsychology experts and native teachers reviewed children’s feedback from this pilot and, with the help of NeuronUP staff, made necessary adjustments to the activities selected for the program to ensure cultural and linguistic adaptation for Guatemalan children.

There are several benefits that the intervention may bring to participants. They will engage in around one hour per week of digital exercises created to stimulate and strengthen cognitive functions and emotional management. These functions are essential for proper child development and have a significant impact on their social and academic performance. It is important to outline the fact that this study will not have a passive control group, as it is part of a bigger international cooperation project conducted by the founding organization focused on providing health and educational services to vulnerable populations, requiring all children to receive some kind of intervention. The control group will use the time dedicated for intervention sessions to carry out a reading motivation program that is regularly used in these schools (a “therapy as usual” control group). This program has been proposed by Fe y Alegría to train reading proficiency while exploring a variety of topics, such as gender equality, sexuality, and many other social topics. They will also perform regular academic activities that the responsible teaching staff consider pertinent. 

This study will contribute to improving the living conditions of Guatemalan children through innovative actions designed to respond to the specific needs of children in vulnerable contexts. Initially, the previous study on socioeconomic status and cognitive development has allowed us to identify the areas that need greater reinforcement [21]. Therefore, the exercises developed using tablets, which have already proven to be successful in other contexts, will be adapted to the needs of the group with which we will work. In addition, the previous study has found anxiety and depression symptoms that are higher than expected and higher than the average, especially in urban–marginal sectors. For this reason, actions to strengthen emotional intelligence are included. Evidence shows that, in addition to improving concentration and academic performance, this type of intervention promotes the prevention of the appearance of violent, addictive behaviours, and other behaviours associated with impulsiveness and difficulty in long-term planning, thus contributing to personal and collective empowerment of these children [27,29,30].

### 2.4. Procedure

The design of this study protocol follows the guidelines of the CONSORT-revised recommendations for improving the quality of reports of parallel group randomized trials [62]. The study has been pre-registered with the Open Science Foundation. Ideally, children should be randomly allocated to either the control or the experimental group. However, implementation of this protocol requires the use of at least two rooms at the same time, along with a high level of logistical organization in terms of moving the children to their correct rooms for each session. Given the nature of the schools involved, we do not believe it is realistic to expect that these resources will always be available. Hence, we propose that the study will be carried out through a multicentre randomized controlled trial in which entire classes of children will be assigned to either the control or experimental conditions. We note, however, that the members of each class are allocated at random by the school as they enroll, so in this sense, random allocation at the participant level is still somewhat achieved. To minimize potential bias related to school-specific factors, each school will provide [at least] two classes, thus ensuring that each school is represented in both conditions. Efforts should be made to ensure as close to a 1:1 ratio of students between groups as possible. The class designated as the experimental group will receive the intervention. The second group will be made up of the students who will form the control group. A crossover design will be used in which the role of each group is reversed during the following academic year, thus ensuring that each child has the opportunity to benefit from the intervention program. The study will therefore be divided into two blocks (see Figure 2). The first block will occur in year 1 and will consist of a prior and posterior assessment (T1 and T2) of the children’s cognitive and emotional abilities and the implementation of the intervention program in between. The second block will occur in year 2 and will consist of a second intervention, followed by a final assessment (T3), which will occur when the children are in the 6th grade. In this block, the control and experimental groups will change roles. Apart from ensuring that all children may benefit from the intervention program, it will provide an opportunity to determine if there is maintenance of the benefits obtained by the experimental group during the first block. All assessments will be carried out by professional neuropsychologists and educational psychologists, duly trained to administer the assessment tasks by the experts from the research team.

The program will take place during the second half of the academic year, at a rate of two sessions per week, with an estimated duration of about 45 min per session. Participants will complete the two weekly sessions on different days of the week, depending on the school they attend, but always with at least one rest day in between. If, for some reason, there are participants who miss one of the sessions, they would have to recover it during the same week to avoid the risk of falling behind their classmates. If participants miss a session and cannot recover it within the same week, they will skip that session and continue with the normal session planned for the following week, but the missed session will be registered as lost. The justification for this decision is that sometimes access to schools or a stable internet connection will not be guaranteed in the context of the study. Hence, this protocol will prevent participants from lagging behind their classmates and avoids having participants completing different sessions in the same classroom. An additional reason for making this decision is that at the end of the program, we will be able to analyze the percentage of completion for each participant and determine the minimum completion percentage to ensure the program’s effectiveness. 

One advantage of the group sessions is that many children can benefit from professional intervention without the need to have one therapist for each child. Nevertheless, children with a previously diagnosed condition are at risk of greater disadvantage compared to their counterparts and may need specialized, one-on-one attention. We do not propose that these children be excluded from the intervention. However, with the help Fe y Alegría, an alternative intervention will be offered to families that wish to receive additional support by referring them to specialized services offered for people with limited economical possibilities. Furthermore, teachers will be provided with training to effectively aid and support these students within the school. 

### 2.5. Data Management and Analysis Plan

All physical materials, such as answer sheets, hand-written responses to the tests, and informed consent originals, will be stored and safeguarded at Fe y Alegría headquarters in Guatemala. The data gathered from the assessments will be transcribed into computer files, which will be stored in secure online servers at Loyola Andalucía University and will only be accessed by authorized personnel from the research project. This data will be compiled and analyzed by the technicians and research assistants collecting the information, supervised by the senior members of the research team functioning as a steering group. Every participant will be assigned an identification number so their data can be compiled and analyzed anonymously, following the Organic Law on Personal Data Protection and Guarantee of Digital Rights 3/2018. If, at the time of data analysis, any potential risks to the participants’ mental health are found by the researchers, an unblinding procedure will take place to identify the participants and notify their school centre so they can take any actions deemed necessary with said children and their families. The funding body will undergo an auditing process in the middle and at the end of the study to ensure that the planned interventions are properly carried out.

Pre/post-evaluation data are best analyzed using an ANCOVA design in which post-scores are the dependent variable, and pre-scores serve as the covariate [63]. This will be conducted separately for each intervention, with the data collected at T2 serving as post-data for intervention 1 and pre-intervention baseline data for intervention 2. If desired, the effects of the two interventions can be directly compared by implementing a repeated measures ANCOVA. Given that it is likely that raw scores will be converted to and analyzed as percentile scores, normal distributions of the dependent variables may not be present. Accordingly, Bootstrapping or non-normal distributions in the Generalized Linear Framework should be considered to accommodate these data. Other covariates, such as sex, age, mean educational level of parents, etc., can be included in the analyses. It will also be possible to perform analyses for different subgroups of participants, either using multilevel, stratified models, or by separating out the subgroups, depending on the specific nature of the subgroups. In particular, we will explore the differences between the “normative” group (participants between the ages of 10 and 12, without any clinical diagnosis and with scores over the inclusion limit in the screening assessment tasks), and the “non-normative” group (participants 13 years old and above, having any clinical diagnosis, or with scores under the screening tasks limit). This last separation will be performed to explore the impact the program may have on children with additional difficulties. We will perform another ANCOVA analysis to explore the long-term changes in the participants at the end of the program (second block), considering the changes they might have experienced on the first block. Where multiple comparisons are undertaken due to the presence of many schools, age ranges, or other factors, appropriate corrections will be applied to the statistical tests. 

Since all children will receive the intervention, we want to explore the effects this program may have on the children who do not meet the inclusion criteria but usually have special educational needs. Due to the nature of the intervention, it is impossible to implement it in a manner that allows both participants and the personnel administering the intervention to participate blindly. However, the personnel from the research team assessing the data will be blinded to group assignment to ensure as much of an unbiased outcome as possible. The interpretation of the data will also have to consider the group assignment to conclude the efficacy of the program.

## 3. Discussion

This study protocol focuses on improving the cognitive and emotional competence of students from especially vulnerable areas of Guatemala City and consequently aims to improve their academic performance and quality of life. In addition, as the participants are minors living in very adverse socioeconomic conditions, they will have the opportunity to learn how to use electronic devices that they would not normally have access to, such as tablets or virtual reality headsets. Given the recent situation with the COVID-19 pandemic, this knowledge can be beneficial for effectively facing the educational challenges that schools currently face after two years of non-regularized schooling. The program is ambitious in terms of the number of tests it proposes to include. Nevertheless, when replicating this study, each research group can choose to vary the number of tests based on their particular research goals and the specific conditions encountered within their sample, and as the intervention progresses.

There is a great educational gap among students caused by the wildly varied situations in their homes. The closure of schools due to the COVID-19 emergency has widened this gap, as most of these vulnerable students do not have access to computers or the internet in their family homes. In addition, the low level of education of their relatives and the fact that they tend to live in small and overcrowded houses have a significant impact on their academic performance. The introduction of tablets has the secondary benefit of making it possible to reduce the digital gap between households while strengthening the capacity for teaching teams to design distance and face-to-face mixed education methodologies that increase the skill levels of students, thus reducing the educational gap. An equitable education is fundamental for exercising rights and increasing opportunities in the long-term.

If the results of this proposed program successfully achieves the expected impact, it could make a big difference in the future of these children, providing them with a chance to develop their cognitive and social abilities to previously unachievable levels, balancing the playing field with other children of their age. This program has been created to be applied in similar low-level socioeconomic contexts, which means that it could become a reliable tool to help children in similar conditions around the world, as the literature suggests [27,29,30]. If the program is effective, there are plans to transfer the knowledge derived from the application of the comprehensive neuropsychological stimulation program in vulnerable areas of Guatemala to particularly disadvantaged contexts in Andalucía (Southern Spain), where 15 of its neighbourhoods are among the poorest in Spain, based on the most recent report of Urban Indicators (Urban Audit) published by the National Institute of Statistics [64].

## 4. Conclusions

The long-term goal of this project is to create a tested and validated cognitive stimulation program, so that it can be integrated into the academic curriculum of educational centres that work with people belonging to this type of population. It is intended to be implemented by the teaching staff of these centres and supervised in parallel by neuropsychology professionals throughout the process. It aims to offer children who live in vulnerable areas the opportunity to receive the support that they may lack during their development in the school phase and allow them to develop to their maximum cognitive and emotional potential. This means that they will be able to increase their academic performance, gain more tools to better manage their mental health and the situations they face living in their communities, and hopefully increase their quality of life.

## Figures and Tables

**Figure 1 healthcare-12-00596-f001:**
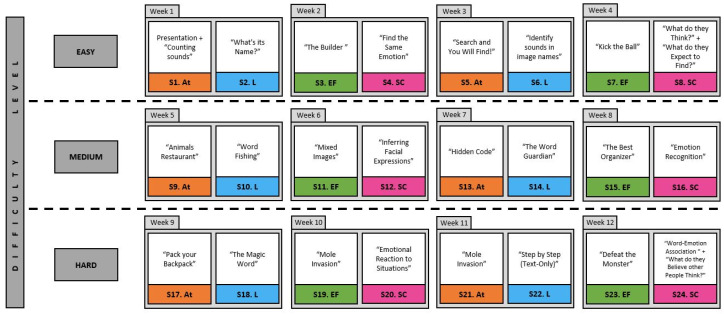
Stimulation program, session by session. Note. The names of every activity in NeuronUp can be seen during each session. In some sessions, participants will engage in more than one activity, depending on the length of each activity. The abbreviations listed alongside the session number refer to: At.: Attention (orange), L.: Language (blue), EF.: Executive Functioning (green), SC.: Social Cognition (pink).

**Figure 2 healthcare-12-00596-f002:**
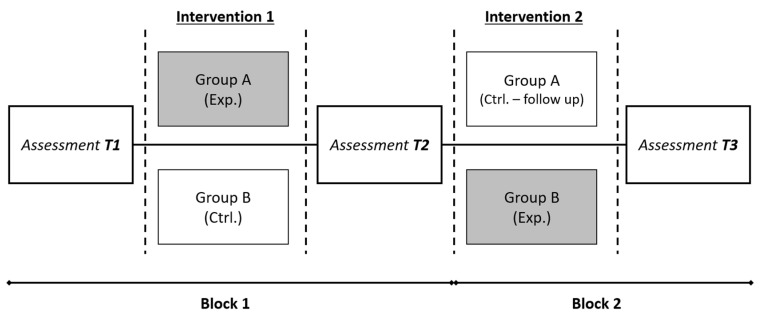
Crossover design of the study. Note. Highlighted in grey boxes are the experimental groups (Exp.) of each round on intervention. White intervention boxes represent the control groups (Ctrl.).

## Data Availability

The datasets generated and/or analyzed during the current study are not publicly available due to the ongoing nature of this study, but if desired, they can be obtained from the corresponding author upon reasonable request. This project was registered in the Open Science Foundation database on the 10 February 2022 (https://doi.org/10.17605/OSF.IO/JVZ6W).

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
