# Peer review of "Neuropsychological Stimulation Program for Children from Low Socioeconomic Backgrounds: Study Protocol for a Randomized Controlled Trial"

_healthcare, 2024, doi:10.3390/healthcare12050596_

Round 1

Reviewer 1 Report

Comments and Suggestions for Authors

Thanks for offering me the opportunity to review this manuscript, which reports the protocol for the evaluation of a neuropsychological stimulation for children from low socioeconomic backgrounds in Guatemala. The researchers are proposing to conduct a RCT and I applause the research team for this initiative. There is a clear justification for the need for such intervention given the several learning, cognitive and psychological issues that children from disadvantage socioeconomic groups in Guatemala are exposed to. However, I would like to offer some suggestions that could help to strength this paper.  

1.     Introduction is well articulated, however it would benefit from a restructure of the presentation of information. There are parts that the reading does not flow well. E.g., by the end of the second paragraph, it is mentioned about the locations of the schools where the intervention will be implemented and tested, however, up to this point there were no information (in the introduction) about the focus of the manuscript. Similarly, in L82 the sentence about ‘creation of the protocol’ comes a bit out of context as, again, there was no information presented beforehand about this manuscript being a protocol. Thus the introduction could be adopt a more linear presentation of facts. Minor points in the introduction are:

2.     L 54: brief say the most common mental health issues associated with covid 19 in Guatemala – poor mental health is a broad term

3.     L 54: add reference(s) for that evidence

4.     L57: add reference to the statement of Guatemala being one of the poorest country in the Americans 

5.     L86: replace ‘creation’ by ‘development’ 

6.     L97: would read better  “This included high levels of anxiety and depression, especially among children from urban areas with high rates of violence (11)” E.g., the sentence is repeating info you already said is affecting mental health so no need to say again about poor quality of life at the end of the sentence.

7.     L 106. Please provide a brief explanation/examples of neurophysiological interventions 

Methods: 

8.     While the tittle of the manuscript states that it is an RCT, it is missing a clear sentence about the design of the study under this section including which type of RCT it is (blinded? Multicentre/single center; parallel group?) This information is briefly mentioned in L258, but should be reported clearly at the beginning of the methods/design study

9.     L157: which evidence was used to select this hight vulnerable neighbourhoods? Add a reference to support it

10.  L173: It is stated that 480 students represent one third of the 5th grade pupils attending  Fe Y Alegria schools. But from the 46 or the 5 selected schools? This is rather unclear. 

11.  L185: replace “detect in recent previous experiences” with “found in our previous studies”

12.  There is a need to provide further information about the intervention. E.g., where the intervention has been developed and tested initially? If not in Guatemala, has it been adapted to that context and with children of the age group that the proposing intervention will target? The cited work of validation studies in other countries are not with children population (line 202). 

13.  How the group allocation will be decided?  In other words, what is the procedure to be adopted? 

14.  L: 242 what means “demarcated inside an international cooperation project” 

15.  L273: Who will training the professionals that will be assessing outcomes? Is this part of the intervention?

Data management and Analysis: This section is far too vague in terms of the analytical plan and must be extended to address the following information:

16.  Who will conduct the analysis? Is there a trial statistician in the team? 

17.  There is no information of which type of statistical analysis will be conducted. Which type of statistical analysis will be conducted for primary and possible secondary outcomes.

18.  Who will be blinded to group allocation? 

19.  Is there a  prespecified analysis plan in place? Is there a trial steering group to foresee the evaluation?

20.  What about data missing? How it will be treated? 

21.  What is the primary outcome and when it will be measured? 

22.  L297: data will be stored in Spain, why? And how it will be transferred to there?

23.  Discussion: is repetitive of the introduction. Suggest rewriting it and make shorter by summarising the steps of the evaluation and just moving into what the intervention can achieve if it shows efficacy and what would be the next steps. 

Comments on the Quality of English Language

The English language is good, however there is a need to revise the way how the information is presented in the introduction as it lacks linearity.

Reviewer 2 Report

Comments and Suggestions for Authors

The manuscript by Rodríguez-Prieto et al. describes a novel and non-standard protocol for conducting a randomized controlled trial of a neuropsychological stimulation program. While the exploration of such a protocol is intriguing, it is advisable to consider conducting at least a pilot study. A pilot study can provide valuable insights into potential implementation challenges, contingency plans, as well as the practical applicability and anticipated outcomes of the proposed protocol.

Due to the thematic focus, it may be more appropriate to submit the work to a specialized psychology journal. Moreover, the manuscript's English should be reviewed by a native language expert. Specifically, colloquial expressions and phrasal verbs should be avoided in scientific English.

Furthermore, subjective statements should generally be avoided in the introduction and throughout the manuscript, except within the discussion section, where interpretations and opinions are appropriately expressed.
The introduction appears to be opinionated and lacks support from references. On the contrary, there are instances where numerous references are appended to support only one statement, such as in line 82.

The introduction, in general, should be more direct, concentrating on the thematic focus of the work.
Additionally, the publication has been deposited in ResearchSquare since July 14th, 2023 (
https://doi.org/10.21203/rs.3.rs-3082159/v1) (https://assets.researchsquare.com/files/rs-3082159/v2_covered_5013c9cd-f633-4a24-9440-2d5886e54c59.pdf?c=1689353909). Therefore, it is necessary for the authors to clarify whether the manuscript is currently under consideration by any other journals.

Minor issues

"emotional" could be added as a keyword

Subjective statements should generally be avoided in the introduction and throughout the manuscript, except within the discussion section, where interpretations and opinions are appropriately expressed.
The introduction appears to be opinionated and lacks support from references.

In the Materials and Methods section, it is recommended to include a detailed description of the participants' socioeconomic data, specifying information such as ages, sex, and other relevant details.

It is possible that the study is overly ambitious in terms of the planned number of participants. Consideration should be given to adopting a more realistic approach or developing a contingency plan to address potential challenges.

The planned number of tests appears to be excessive, and as a result, adherence to them may be low, leading to incomplete test submissions. Consider revising the testing plan to ensure practical feasibility and optimal participant engagement.

It is advisable to incorporate a teaching component to instruct participants on the proper usage of tablets. This can enhance their familiarity with the technology and contribute to improved engagement and data quality.

The description of Figure 1 in both the text and the figure itself is inadequate. Additionally, the letter size in the figure is excessively small and may impede clarity. Consider revising the description for better comprehension and adjusting the letter size for improved visibility.

The potential heterogeneity within the sample could pose challenges for interpreting the results. It is recommended to consider stratifying the sample to account for these variations and enhance the robustness of the interpretation. The utilization of multiple centers, diverse age groups, and both sexes might introduce heterogeneity that should be addressed through statistical correction methods (multiple tests correction).

The presence and, if applicable, the nature of controls are not clearly articulated. Clarification on the inclusion of controls, along with their type, is necessary for a comprehensive understanding of the study design. In Figure 2, it appears that in Intervention 2, a treated group is utilized as the control.

 Spanish laws are applied in Guatemala (line 302). Is this legally correct?

Lines 308 to 324 introduce an analysis of the results that should be considered before implementation, not after. For the group with issues not solvable with this intervention, alternative interventions should be contemplated.

In the discussion, it is advisable to avoid using the COVID-19 pandemic as a justification for the use of tablets, as current circumstances may not involve confinement as a viable option.

The paragraph from line 342 to 355 appears to be more closely related to the Materials and Methods section. Consider relocating this content to enhance the logical flow and coherence of the manuscript.
At the end of the discussion, the authors mention the expected impact and results but do not provide information on how to evaluate these outcomes. Consider addressing this aspect to ensure a comprehensive understanding of the anticipated impact.
If the last paragraph serves as a conclusion, consider titling it as such for clarity and structure.

Reviewer 3 Report

Comments and Suggestions for Authors

The authors outline a promising study.

1. This study aims to improve psychological development in school-aged children in Guatemala, a country with high rates of poverty and socioeconomic inequalities, putting children at risk for their development.

2. The study's procedures seem promising as they recruit and try to include all children through recruitment and intervention at schools in vulnerable and underprivileged areas of Guatemala City.

3. The study is rather an implementation / intervention study than providing new knowledge. Its strength is that it aims to strengthen children at risk and thus to increase equity. Also, it will test for possible intervention effects, thus it is an efficacy trial for the developmental program they describe. It adds knowledge about efficacy in a new population - the introduction and description of the population and their specific vulnerabilities is profound and concise.

4. The authors should consider an additional measurement as a follow-up 6 or 12 months after the end of the intervention to test for long-term effects (T3).

5. It is a research protocol, so no data or results yet to interpret or discuss

Reviewer 4 Report

Comments and Suggestions for Authors

Dear authors,

It is a work of protocol with interesting data relating to its structure; in addition to this, it is recommended that the following suggestions be taken into account in order to strengthen its quality. 

The abstract should be concise and clear, it is recommended that this section should be revised to follow the IMRD structure. 

There is no solid and pertinent discourse to justify this protocol. In particular, the literature review does not focus on the emotional and cognitive factors studied in depth. 

There are sections in the literature review that are more appropriate for discussion than this section. There is also information repeated throughout the document that is redundant. 

The information on line 119 is unclear, please rephrase this sentence. 

Section 2.3.1 mentions a battery of tests used to obtain information but does not include the degree of reliability of these tests. 

Also, the current wording does not allow for a discussion that highlights the relevance of the research findings with work by other authors that supports this protocol work. 

Double spaced throughout the document 

Review of references in accordance with journal standards.

Best regards

Reviewer 5 Report

Comments and Suggestions for Authors

Thank you for the opportunity to review this interesting paper. I recognize the social impact of this study; therefore, it is crucial to provide many revisions. My comments primarily focus on methodological procedure.

1.     Theoretical background

A national statistic that supports the need to implement the work proposed in the study could be included by the authors.

2.     Participants and study setting

Suggestion: You split the sample description.  First, you describe would like to do, the characteristics, secondarily G power size: I think is important to revise the structure of this section. For example: G power indication, motivation about you probably will recruit less participants, and the characteristics. It looks confused.

Row 130: “Students from 5th grade of primary school, from educational centers located in..”. Considered the complicated social background of you sample could be important to specify the participants’ age. All your participants will have the same age of some of them because of social difficulties could be older than other?

Statistical analysis

3.     The study protocol needs to include the statistical part; it does not explain in detail the analyses that need to be performed. For instance, it does not describe how to compare test scores to different scales and whether they should be transformed into Z scores. Moreover, the authors should mention the software to be used for the analysis. They should perform the data analysis using open-source software.

Comments on the Quality of English Language

Minor editing of English language required. 

Round 2

Reviewer 1 Report

Comments and Suggestions for Authors

The authors addressed all my points thoroughly. No further comments from me. Well done and good luck in running the trial.

Author Response

Thank you for all your constructive feedback and helping us perfect our article. We really appreciate the advice from peers like you who want to help out other professionals.

Reviewer 2 Report

Comments and Suggestions for Authors

Some major and minor issues have not been addressed in the reviewed version. Therefore, I recommend accepting the document after a major revision.

-It is recommended that the manuscript's English be reviewed by a native language expert. One major issue is the use of colloquial expressions and phrasal verbs in scientific English.

- Subjective statements should generally be avoided in the introduction and throughout the manuscript, except within the discussion section, where interpretations and opinions are appropriately expressed. 

-The introduction appears to be opinionated and lacks support from references. On the contrary, there are instances where numerous references are appended to support only one statement, such as in line 82. The introduction should focus on the main theme of the work and be more direct.

-Consider creating a contingency plan to address any potential challenges related to sample and data collection.

-The current number of tests may be excessive, leading to incomplete test submissions due to low adherence.  Consider revising the testing plan to ensure practical feasibility and optimal participant engagement.

-The study design lacks clarity regarding the presence and type of controls.  It is necessary to clarify the inclusion of controls and their nature to ensure a comprehensive understanding. Figure 2 suggests that a treated group is used as the control in Intervention 2.
-To improve the robustness of the interpretation, it is recommended to consider stratifying the sample to account for potential heterogeneity. Statistical correction methods (multiple tests correction) should be used to address any heterogeneity introduced by the utilization of multiple centers, diverse age groups, and both sexes.

Comments on the Quality of English Language

-The English of the manuscript should be checked by a native speaker. In particular, colloquial expressions and phrasal verbs in scientific English should be avoided.
